# Molecular Diagnosis of Endemic Mycoses

**DOI:** 10.3390/jof9010059

**Published:** 2022-12-30

**Authors:** Clara Valero, María Teresa Martín-Gómez, María José Buitrago

**Affiliations:** 1Manchester Fungal Infection Group, Division of Evolution, Infection, and Genomics, Faculty of Biology, Medicine and Health, University of Manchester, Manchester M13 9WU, UK; 2Microbiology Department, Hospital Universitari Vall d’Hebron, 08035 Barcelona, Spain; 3Mycology Reference Laboratory, Centro Nacional de Microbiología, Instituto de Salud Carlos III, Carretera Majadahonda-Pozuelo Km 2, Majadahonda, 28220 Madrid, Spain; 4CIBERINFEC, ISCIII-CIBER de Enfermedades Infecciosas, Instituto de Salud Carlos III, Majadahonda, 28220 Madrid, Spain

**Keywords:** endemic mycoses, molecular diagnosis, PCR, NGS

## Abstract

Diagnosis of endemic mycoses is still challenging. The moderated availability of reliable diagnostic methods, the lack of clinical suspicion out of endemic areas and the limitations of conventional techniques result in a late diagnosis that, in turn, delays the implementation of the correct antifungal therapy. In recent years, molecular methods have emerged as promising tools for the rapid diagnosis of endemic mycoses. However, the absence of a consensus among laboratories and the reduced availability of commercial tests compromises the diagnostic effectiveness of these methods. In this review, we summarize the advantages and limitations of molecular methods for the diagnosis of endemic mycoses.

## 1. Introduction

The common term “endemic fungi” usually refers to fungal species within the Onygenales order sharing, among others, four distinctive features: (i) thermal dimorphism, (ii) geographical distribution restricted to specific regions of the world, (iii) ability to cause a disease in otherwise healthy humans, although illness tends to be more severe in immunocompromised individuals, and (iv) high mortality rates if the illness fails to be timely diagnosed and treated [1]. Recently, the WHO has released the fungal priority pathogen list to strengthen the global response to fungal infections. Several endemic fungi are listed within the high and medium priority groups [2].

### 1.1. Epidemiology of Endemic Mycoses

Endemic mycoses (EM) are caused by species of the genera *Histoplasma*, *Blastomyces*, *Coccidioides*, *Paracoccidioides*, *Talaromyces*, *Sporothrix*, *Lacazia,* and the recently described *Emergomyces*. The distribution area of endemic cases encompasses countries across the five continents. Coccidiomycosis, paracoccidiomycosis and lobomycosis are restricted to the American continent, whereas sporothrichosis and histoplasmosis have a cosmopolite distribution with high presence in the Americas and Africa. Blastomycosis extends mainly across Africa, the western basins of United States of America (USA), and the south-western Canadian border. Talaromycosis cases are typically found in south-eastern Asia, while emergomycosis is frequently diagnosed in South Africa, but cases have also been reported in North America, Europe, Asia and India [3,4]. Certain species within endemic genera can be found only in specific areas of the world, usually associated to particular environmental conditions of heat, moisture, pH or nutrients, among others [5].

The true epidemiology of endemic fungal infections is unknown. Many primary infections are asymptomatic or present with mild self-resolving symptoms not requiring the search of medical care, and frequently the etiological agent of the infection fails to be identified due to lack of awareness and limited access to the appropriate diagnostic tools. This is particularly concerning outside hyperendemic territories, where the vast majority of EM cases are imported and associated to immigration and travels from endemic areas [6]. Despite EM global burden is increasing, clinical infections are not subjected to mandatory notification to Public Health systems with exceptions restricted to specific areas [7,8].

The incidence of histoplasmosis has been estimated to range between 0.1–100 cases/100,000 inhabitants, with lowest rates observed in areas with temperate climates, and the highest incidence in humid tropical territories [9]. Serologic studies indicate that up to 40% of the population living in highly endemic areas may have been exposed to the fungus, with seropositivity reaching up to 87% in specific populations [7]. The real number of coccidioidomycosis cases has been estimated to exceed 350,000 per year in the USA, with an increasing trend observed over the last years [10,11,12]. In Brazil, paracoccidioidomycosis is estimated to affect 3–4 new patients/100,000 inhabitants/year, with an incidence that may reach up to 40 patients/100,000 inhabitants depending on the location. In this country, paracoccidioidomycosis represents the main cause of hospitalization and death among the overall systemic mycoses [3]. Talaromycosis is one of the most neglected and underrecognized EM as its prevalence is largely unknown. This disease is strongly associated to poverty and uncontrolled advanced HIV disease, especially in areas where the access to healthcare is limited. Some reports describe that the burden of this disease could exceed 17,000 cases/year, being lethal in as much as 1 in 3 cases [13]. With approximately 40,000 new cases every year sporotrichosis is considered the most prevalent EM in South America, and the most frequent EM in regions of Southern Brazil [14]. It is also endemic in Mexico and Northern China, and has been responsible of large outbreaks in North-America, Australia, and South Africa [15]. Emergomycosis has been described as an HIV-associated infection in South Africa, where it ranked the second most frequent EM only after sporotrichosis in a recent review of contemporary cases spanning 10 years [16]. Scattered reports locate *Emergomyces* spp. also in Europe, North America, and Asia. The true incidence of emergomycosis is unknown, but after the introduction of molecular techniques, many cases initially classified as histoplasmosis on the basis of histopathology have been demonstrated to be emergomycosis, indicating that its prevalence may be more frequent than previously thought [17]. Lobomycosis prevalence is unknown but an increase in new cases has been observed in recent years [18].

EM incidence has been reported to be on the rise in recent years [3,7,19,20]. This has been mainly attributed to environmental changes, travels, and expansion of at-risk populations, along with increased awareness and wider access to improved diagnostic techniques. As more research and educational actions are undertaken, new areas devoted to combatting these diseases will be uncovered [7,13,21,22]. Thus, a wide availability of sensitive, specific, rapid, and versatile diagnostic techniques will become an immediate necessity.

### 1.2. Diagnosis of Endemic Mycoses

To date, the laboratory diagnosis of EM is an unsolved issue [23]. The diagnostic yield of currently available microbiological techniques has been extensively reviewed by an international team of experts in a joint ECMM-ISHAM initiative, resulting in evidence-based recommendations of use recently published [4]. Conventional techniques, such as histopathology and culture, are not difficult to implement, but exhibit a number of limitations that should be taken into consideration. Firstly, these techniques require a high level of expertise and special caution is needed when handling specimens and cultures, as some species are classified as BSL-3 microorganisms; depending on the specimen and phase of the illness [24]. Moreover, cultures may delay the diagnosis up to four-to-six weeks, as these fungal species are slow-growing, and the confirmation of the dimorphism may be required for the final identification. In addition, their diagnostic yield is hampered by lack of sensitivity, particularly in non-invasive chronic forms.

Culture independent commercial assays, which rely on the detection of antigens or antibodies in clinical samples, are only available for the most prevalent EM. Antibody-based diagnosis is determined by the immune status of the host, as immunosuppresed patients fail to produce high antibody titers and seropositivity remains long time after the infection [25]. Moreover, these tests exhibit cross-reactivity among EM-causing species and with other human fungal pathogens. Antigen tests have been proved to be useful for rapid diagnosis in populations generally affected by severe immunosuppresion and disseminated forms of disease [26], but little information is available on their applications in other contexts. Specificity of antigen tests is reduced by cross-reactivity issues with other fungi [27]. During the diagnostic process, the possibility of cross-reactivity of antigens and antibodies shall be considered in areas where endemic genera co-exist. Point-of-care methods have been developed for the detection of *Coccidioides* spp. and *H. capsulatum*, although studies to date are limited, results seem to be promising [28,29].

Molecular techniques have been key for taxonomic placement and to uncover cryptic species withing the EM-causing species [30], but their application to clinical diagnosis is far from being of routine use. Most specific PCR techniques have been developed by reference laboratories without a consensus about the technology used (conventional PCR, quantitative PCR, LAMP etc.) or the genomic regions targeted by the assays. Despite several techniques for the detection of EM have been reported to be useful for diagnosis, only a *Coccidioides*-specific PCR is commercially available ((accessed on 28 December 2022)).

Despite the efforts on validating molecular assays in diverse types of patients and samples, the limited presence of molecular techniques in international diagnostic guidelines is due to the need of further standardization, and the lack of solid multicentered studies involving large populations. Recently, an European initiative has established a working group devoted to perform intercomparison multicenter diagnostic studies with the objective of improving EM diagnosis and acquiring a better knowledge about the epidemiology of these neglected fungal infections (https://www.ecmm.info/working-groups/working-group-on-the-diagnosis-and-the-epidemiology-of-endemic-mycoses (accessed on 28 December 2022)). Notwithstanding these limitations, molecular techniques currently seem to represent a good immediate alternative for a fast and specific diagnosis of such infections, as well as a feasible tool to go deeper into the knowledge of their epidemiology [31].

This review is intended to summarize the techniques, targets, applications of molecular techniques to the diagnosis of endemic mycoses, covering the full spectrum of techniques, from the most traditional PCR protocols to the most advanced sequencing methods (Figure 1).

## 2. Specific PCR Assays

Specific PCR assays have been developed last years in reference laboratories mainly focused on the detection of *H. capsulatum* and *Coccidioides* spp. For the remaining EM species, there are considerably fewer studies. In general, commercial tests and inter-comparison studies are lacking. The global SARS-CoV-2 pandemic has allowed the implementation of conventional and real-time PCR (qPCR) technology in several laboratories worldwide, including endemic regions, which offers an excellent opportunity to expand the application of molecular techniques for the detection of these neglected pathogens in a near future.

### 2.1. Histoplasmosis

Methods based on PCR (conventional or real time) for the detection of *H. capsulatum* target different genomic regions: (i) ribosomal DNA (rDNA) multicopy regions as 18S [32], ITS1 and ITS2 regions [33,34,35,36] and the ribosomal small subunit RNA [37], or (ii) unicopy targets as genes coding the 100-kDa-like protein or the M antigen [38,39,40,41,42] and, more recently, *PPK* and *CFP4 genes* [43].

DNA from clinical and reference isolates or/and clinical samples has been used for validation of these specific PCR assays. The type of clinical samples varies including respiratory secretions, biopsies, bone marrow, blood, or sera. In general, better sensitivity values were reported using clinical specimens sampled at the site of the infection, such as respiratory and biopsy samples. However, less invasive samples, such as sera and blood, were often preferred in disseminated infections [33,37,41]. Methods for the nucleic acid extraction from clinical samples also differed depending on the assay with some including sample pretreatment and others using total nucleic acids, the latter introducing a reverse transcription step before the amplification of the target to improve sensitivity [37]. Although the number of clinical samples in some publications was very limited, sensitivity values reported in these studies ranged from 70–100% [35,38]. A recent meta-analysis focused on HIV+ patients with progressive disseminated histoplasmosis reported an overall sensitivity and specificity of (95% CI) of 95.4% (88.8–101.9) and 98.7% (95.7–101.7), respectively, in different type of samples including respiratory, biopsies, blood and bone marrow [44].

LAMP methods described for the diagnosis of histoplamosis are scarce. These assays have been designed to target the ITS region [45] or the 100-kDa-like protein [46] showing variable sensitivity results.

Regarding the establishment of a consensus about histoplasmosis PCR diagnostic methods, to date, only one multicenter study involving laboratories from four Latin American countries and Spain has been published [47]. In this work, seven different PCR protocols were compared using the same DNA panel for testing the assays. Although the overall sensitivity and specificity was 86 and 100%, respectively, PCR real-time based protocols were demonstrated to be the most sensitive and reproducible approaches compared to conventional PCR assays. Methods targeting unicopy genes showed the poorest sensitivity.

### 2.2. Coccidiomycosis

Molecular techniques for the detection of *Coccidioides* spp. have been developed to be used on both clinical [48,49] and environmental settings, such as endemic regions from USA, where a steady rise in coccidioidomycosis infections has been reported [50,51]. These assays were designed to target the ITS region of the ribosomal DNA and genes encoding both Antigen 2 and Proline rich Antigen, with sensitivity ranging from 74 to 100%. Clinical samples used to test these assays were mainly respiratory, fresh and paraffin embedded biopsies and cerebrospinal fluid. When comparing different clinical samples, the best performance was obtained when using respiratory samples, fresh tissues reached 93% sensitivity, and paraffin-embedded tissues sensitivity was reported to be around 73% [48]. In 2018, the FDA authorized a commercial assay for the rapid detection of coccidioidomycosis, the Genestat MDX *Coccidioides* (https://www.aacc.org/cln/articles/2018/march/fda-clears-first-molecular-test-for-valley-fever (accessed on 28 December 2022)). In a multicenter study, this method reached a 100% sensitivity, with a specificity that ranged between 93.8% and 100% depending on the sample tested [52].

Of interest, *Coccidioides* spp. is the only fungal genus included in the international lists of potential bioterrorism agents [53], making essential to be able to face this contingency with the aid of a rapid detection method. In this line, molecular techniques represent an excellent option to be included in preparedness and response protocols due to their short turnaround response time and remarkable sensitivity and specificity scores. However, further standardization and consensus are needed.

### 2.3. Paracoccidioidomycosis

Several “in house” molecular techniques have been described for the detection of *Paracoccidioides* spp., especially in laboratories from Brazil and other non-endemic regions (Table 1). Most of these assays were based on conventional PCR methodologies [54,55,56,57]. On the other hand, two methods based on qPCR for their use in paracoccidioidomycosis diagnosis have been published [31,58]. Targets selected for amplification were the multicopy ITS rDNA region and the genes encoding the proteins Gp43 or Pb27. The clinical samples tested in theses assays were mainly respiratory, biopsies, blood and sera. Of note, sera samples were not recommended in two of these studies [31,56] as authors never detected DNA in these kinds of samples. The overall sensitivity ranges reported were 91–100%, showing a great potential of these techniques for clinical use.

Only one LAMP method has been described to date targeting the gene encoding the Gp43 protein; however, the sensitivity reported on sputum samples was moderate (61%) [59].

*Paracoccidioides* spp. are considered fastidious microorganisms as recovering these pathogens from culture is hard and time-consuming, commercial antigen tests are still not available and serological methods have strong limitations. Considering all the above, the inclusion of these molecular diagnostic techniques in the routine of clinical microbiology laboratories is substantially justified. This is even more imperative in non-endemic regions where the delay in diagnosis has fatal consequences in paracoccidioidomycosis patients [60,61,62].

### 2.4. Blastomycosis

Fewer assays have been described for the diagnosis of blastomycosis. The *BAD1* gene, an important conserved adhesion-promoting protein and virulence factor of *Blastomyces* spp. has been chosen as target in several assays developed for the detection of the fungus in soil [63] or in clinical samples [64]. Other targets as *DRK1* gene have also been used [65]. Although there is little evidence of the usefulness of theses assays in a clinical setting, the results obtained were very promising with high specificity and sensitivity values reported.

### 2.5. Talaromycosis

A recent meta-analysis has reviewed the methods based on PCR developed for the rapid diagnosis of talaromycosis [66]. Most of them have been published by authors from endemic regions (China, Vietnam, Thailand) which used conventional nested PCR [67] or real-time PCR [68,69] targeting the ribosomal DNA or other gene encoding regions [70]. Samples tested included plasma, blood, serum and bone marrow reporting an overall sensitivity and specificity of 84% and 99%, respectively. A LAMP assay has been published recently showing a suitable sensitivity and detecting all the biopsy samples tested [71].

### 2.6. Conclusions

Although data are very heterogeneous among works, specific PCR assays are rapid sensitive and specifics. Some studies used a limited number of samples for the validation of the assays, and studies focus on blastomycosis and talaromycosis are scarce. Reaching consensus about targets and kind of samples should be a priority (Table 1).

**Table 1 jof-09-00059-t001:** Details of the studies where specific PCR assays were used to diagnose endemic mycoses.

PCR Technology	Target	Sample	Sensitivity (Cases)/Specificity	Specificity	Ref
	** *Histoplasmosis* **				
Conventional (nested)	18S rDNA	Blood, spleen, lung (mice)	83.1%	ND	[32]
Conventional (nested)	100-kDa-like protein gene	Biopsy	70%	100%	[72]
Conventional	M antigen gene	ND	100%	100%	[39]
Conventional (semi-nested)	M antigen gene	Biopsy, blood, mucose, BM	ND (30)	ND	[38]
Real-time	ITS rDNA	BAL, lung biopsy, BM	100% (3)	100%	[35]
Conventional (nested)	100-kDa-like protein gene	Blood, serum, BAL, BAS, biopsy, CSF, others	100% (40)	100%	[41]
Real-time	ITS rDNA	Blood, serum, BM, sputum, BAS, BAL, biopsy, CSF, others	89% Proven H (54) 60% Probable H (13)	100%	[31]
Real-time	ITS rDNA	BAL, biopsy, BM, CSF	95.4% (348)	96%	[36]
Real-time (multiplex)	ITS rDNA	BAL, biopsy, serum, BM	92.5% (72)	100%	[34]
Real-time	*mtSSU* gene	Blood, serum, BAL, BAS, biopsy, CSF, others	97.7% (44)	ND	[37]
Conventional Real-time	*PPK, CFP4*	FFPE tissue	100% (2)	ND	[43]
	** *Paracoccidioidomycosis* **				
Conventional (nested)	Gp43	Biopsy (mice)	91% (23)	ND	[57]
LAMP	Gp43	Sputum	60% (18)	ND	[59]
Conventional (semi-nested)	ITS rDNA	Biopsy (mice)	100% (4)	100%	[54]
Real-time	ITS rDNA	Serum, blood, sputum	100% (6)	ND	[73]
Conventional	ITS rDNA	Serum, biopsy	ND	ND	[56]
Conventional (semi-nested)	ITS rDNA	Sputum	100% (14)	ND	[74]
Conventional (nested)	GP43 gene	BAL, biopsy, sputum	100% (25)	100%	[55]
Real-time	Pb27 gene	Blood, serum, biopsy and others	94% (78)	100%	[58]
	** *Coccidioidomycosis* **				
Conventional (nested)/real-time	Antigen2/Proline-Rich Antigen,	FFPE- biopsy	100% (3)	ND	[75]
Real-time	ITS rDNA	Respiratory, biopsy, FFPE-biopsy	89% (480)	98%	[48]
Real-time	ITS rDNA	Mice samples	98% (44)	100%	[49]
Real-time	GeneSTAT *Coccidioides* assay	BAL/BW	100% (332)	93.85–100%	[52]
	** *Blastomycosis* **				
Conventional (nested)	*WI-1* (*BAD 1*)	PE-biopsy (dogs)	ND (73)	ND	[76]
Real-time	*DRK-1*	Respiratory, biopsy and others	86% (14)	99.4%	[65]
Real-time	*BAD-1*	FFPE-biopsy	83% (12)	100%	[64]
Real-time (duplex)	*BAD-1*	FFPE-biopsy, respiratory and others	ND (33)	ND	[77]
	** *Talaromycosis* **				
Real-time	5.8S rDNA	Blood	60% (20)	100%	[78]
Conventional (nested)	18S rDNA	Serum	68.6% (35)	100%	[67]
LAMP	ITS rDNA	Biopsy	100% (12)	100%	[71]
Conventional (nested)/ real-time	ITS rDNA	Blood, serum	82% (22)/91% (22)	75%/63%	[68]
Real-time	ITS rDNA	Serum	86.11% (36)	ND	[69]

ND: no data; FFPE-biopsy: formalin-fixed paraffin-embedded biopsy; BAL: brochoalveolar lavage; BAS: brochoaspirate; BW: bronchial wash; CSF: cerebrospinal fluid; BM: bone marrow.

## 3. Broad-Range PCRs

Broad-range or panfungal PCR assays are especially useful for EM diagnosis, generally used when there is not a clear suspicion of the fungal agent causing the disease, which is one of the hallmarks of EM, or when the infection is not frequent in the setting of the diagnostic laboratory, as it is in non-endemic areas. This approach relies on the use of fungal (or fungal group)-specific primers to amplify fungal DNA directly from clinical samples followed by an identification method, mainly Sanger sequencing, to confirm the causative agent [79]. With the aim of improving sensitivity, classic multi-copy targets as the ribosomal operon [37] are often selected for panfungal amplification, while fresh tissue samples are preferred over formalin-fixed, paraffin-embedded samples [80].

Sample contamination, detection of commensal fungi, PCR bias due to primer mismatches and, the lack of adequate reference databases for fungi identification are the main limitations of panfungal PCR assays. However, the limitation of delay in response time associated to species determination has been addressed by replacing Sanger sequencing identification with other time-saving post-PCR methods such as melting curve analysis, DNA microarray, electrospray-ionization mass spectrometry analysis and T2 magnetic resonance [81].

In conclusion, although proper studies directed to EM diagnosis by using broad-range PCRs are still missing, there are plenty reports in the literature showing the ability of this technique to provide a definite diagnosis when paired with other reference methods. This technique has the advantage of being cost-effective and can be an alternative to specific PCR considering their limitations (Table 2).

## 4. Next Generation Sequencing (NGS)

NGS has revolutionized the diagnosis of fungal and other microbial infections and it is already considered the future replacement for the current broad-range PCR methods. The most used NGS approach for diagnosis nowadays is targeted amplicon sequencing or metabarcoding. By using fungal-specific primers, thousands of copies of different DNA templates are amplified and sequenced simultaneously, reducing turnaround time and costs [95]. However, shotgun metagenomic sequencing can be also used to target most parts of the genomes of the microorganisms present in the sample. This approach is more expensive and computationally demanding, but allows for further characterization of the infecting agent as other features, such as identifying the subtype or the antimicrobial resistance profile, could be retrieved from the sequenced data [96]. In general, NGS methods face the same limitations as broad-range PCR assays but with the additional requirement of expertise in data analysis and increasing complexity in the technical procedures [97].

NGS technologies were originally standardized as an exploratory tool to study the fungal community profile (mycobiome) of human specimens. As an example, McTaggart LR and colleagues developed an NGS-based method for the analysis of the lung mycobiome during *Blastomyces dermatitidis/gilchristii* infection [98]. The successful detection of the causative agent as well as other fungal pathogens indicated the potential of this method for the diagnosis of EM. However, proper standardization and retrospective studies including a substantial number of clinical isolates are still missing, it not being currently possible to recommend or suggest a method or to consider NGS as a suitable tool for EM diagnosis. Most studies reported in the literature describe brief case reports or anecdotical presence of EM samples in bigger specimen sets. Nevertheless, NGS methods have already been employed successfully in the differential diagnosis of infections with similar clinical symptoms and the identification of the biological source of an outbreak (Table 3). Recently, the assessment of the clinical performance of NGS for the rapid diagnosis of talaromycosis in HIV patients has been evaluated [99]. The sensitivity of the new method was significantly higher than culture and serum galactomannan determination (98.3% vs. 66.7% and 83.3%, respectively) underlining the potential use of NGS for EM diagnosis. In conclusion, although the NGS-based method seems to be promising, more studies need to be able of consider it as a tool for the diagnosis of EM (Table 3).

## 5. Conclusions and Perspectives

Diagnosis of EM is still difficult in endemic regions and even more complicated out of these regions, where the lack of suspicion and expertise are the major shortcomings. Molecular techniques have shown their great potential for the rapid diagnosis of EM in several studies performed in reference laboratories in the last years. The recent COVID-19 pandemic has not only increased the awareness on how critical a rapid diagnosis is but paved the way to the generalized implementation of the molecular diagnosis of infectious diseases. As summarized in this review, several molecular techniques developed in recent years show a great potential for the rapid diagnosis of EM. In non-endemic countries, where the availability of some other useful techniques, as antigen detection, is limited, qPCR-based molecular assays have been developed to this purpose, extending their usefulness to difficult-to-diagnose forms of infection [34,37]. The introduction of multiplex formats also allows for performing a differential diagnosis with other pathogens causing similar clinical patterns reducing costs [118]. In endemic areas, especially in resource-limited settings, cost-effective molecular methods such as LAMP could be a promising alternative. However, in general terms, there is still great variability in published methods to date and commercial kits are practically non-existent. An effort to standardize and achieve a consensus should be performed among the different laboratories. Technical issues such as the selection of genomic targets or nucleic acid extraction methods, coupled with the implementation of inter-comparison studies should be prioritized to include these techniques in the future guidelines for patient management. Panfungal assays stand for an interesting alternative to specific assays as these techniques are easy to implement and more cost-effective; however, limitations of these tests should be considered when performing a final diagnosis. Recently, NGS has emerged as an alternative to overcome some of these limitations soon. As a conclusion, the implementation of molecular techniques in clinical settings will revolutionize the rapid diagnosis of EM, especially in countries where laboratories use diagnostic PCR routinely.

## Figures and Tables

**Figure 1 jof-09-00059-f001:**
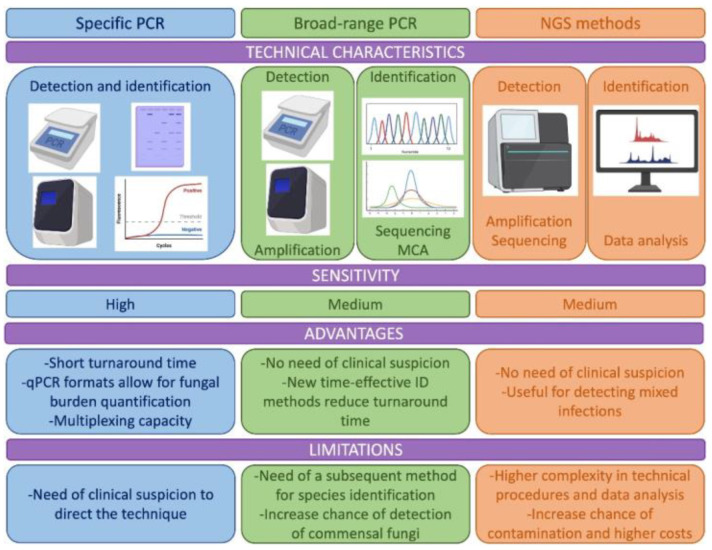
Description of the technical characteristics, advantages and limitations of the molecular methods used for the diagnosis of endemic mycoses including specific PCR methods, methods based on broad range PCR and new methods as those based on Next Generation Sequencing (NGS). MCA: melting curve analysis.

**Table 2 jof-09-00059-t002:** Details of the studies where broad-range PCR was used to diagnose endemic mycoses.

Target	Sample	Post-PCR ID Method	Notes	Ref
** *Histoplasmosis* **				
rDNA (18S)	BM	Sanger sequencing	Confirmed by histopathology and culture	[82]
rDNA (ITS1)	BM	Sanger sequencing	Confirmed by culture	[83]
rDNA (ITS, 28S)	Lung tissue	Sanger sequencing	Confirmed by histopathology	[84]
rDNA (28S)	Mucosal biopsy	Sanger sequencing	Confirmed by specific PCR	[85]
rDNA (28S)	FFPE tissue	Sanger sequencing	Confirmed by histopathology and specific qPCR	[86]
** *Coccidioidomycosis* **				
rDNA (ITS)	Biopsy	Sanger sequencing	Confirmed by histopathology, qPCR format	[87]
** *Blastomycosis* **				
rDNA (ITS2 and D2)	FFPE tissue	Sanger sequencing	Confirmed by histopathology	[88]
** *Emergomycosis* **				
rDNA (28S, ITS2)	FFPE tissue	Sanger sequencing	Confirmed by histopathology	[89]
** *Lobomycosis* **				
rDNA (ITS1-4)	Biopsy	Sanger sequencing	Confirmed by histopathology	[90]
** *Multiple EM identified* **				
rDNA (ITS2)	Biopsies	MCA and sanger sequencing	Histoplasmosis, coccidioidomycosis, paracoccidioidomycosis. Confirmed by histopathology	[91]
rDNA (28S, ITS2, D1-D2)	FFPE and fresh tissue	Sanger sequencing	Histoplasmosis, talaromycosis, blastomycosis. Some cases confirmed by histopathology	[92]
rDNA (ITS2, D2)	FFPE tissue	Sanger sequencing	Histoplasmosis, coccidioidomycosis. Confirmed by histopathology, qPCR format	[93]
rDNA (ITS1-2)	FFPE and fresh tissue	Sanger sequencing	Histoplasmosis, paracoccidioidomycosis. Confirmed by culture or histopathology	[94]

BM: bone marrow; FFPE: formalin-fixed paraffin-embedded; MCA: melting curve analysis.

**Table 3 jof-09-00059-t003:** Details of the studies where NGS was used to diagnose endemic mycoses.

Target	Samples	Aim	Notes	Ref
** *Talaromycosis* **				
Total DNA	BAL, CSF and BM	Diagnosis of a patient with a 3-months record of undiagnosed disease	Confirmed by histopathology and positive culture in skin lesion	[100]
Total DNA	CSF	Diagnosis of a patient with meningoencephalitis		[101]
Not mentioned	BAL	Diagnosis of a patient with chronic pneumonia	Confirmed by culture in BAL	[102]
Total DNA	Peripheral blood	Diagnosis of HIV febrile patient	Confirmed by panfungal PCR on lymph node biopsy	[103]
Not mentioned	BAL	Diagnosis of a patient with chronic pneumonia	Confirmed by culture in BAL	[104]
Total DNA	Skin tissue and eye aqueous humor	Diagnosis of a patient with eye tumor	Confirmed by PCR in the aqueous humor	[105]
Not mentioned	BAL and blood	Diagnosis of a patient with chronic pneumonia	Confirmed by culture in sputum	[106]
Total DNA	FFPE tissue	Differential diagnosis of a patient with peritonitis		[107]
Not mentioned	BAL	Diagnosis of a patient with chronic pneumonia	Confirmed by culture in BAL	[108]
Total DNA	BAL, blood, and BM	Assessment of clinical performance of NGS for talaromycosis diagnosis	Sensitivity and specificity values were 98.3 and 98.6%, respectively. The clinical final diagnosis was used as the reference standard.	[99]
** *Histoplasmosis* **				
Total RNA	CSF	Differential diagnosis of meningitis	Statistical framework supported by environmental and non-infected control samples	[109]
Total DNA	Miscellaneous	Identification of the causative agent causing an outbreak		[110]
Not mentioned	Not mentioned	Diagnosis of a patient with chronic progressive lung lesions		[111]
DNA (ITS region)	FFPE tissue	Diagnosis of a patient with a skin lesion	Confirmed by histopathology	[112]
Not mentioned	BM	Diagnosis of non-HIV febrile patient	Confirmed by direct visualization	[113]
** *Blastomycosis* **				
Cell-free DNA	Plasma	Diagnosis of a patient with chronic pneumonia		[114]
Not mentioned	BAL and biopsy	Diagnosis of a patient with chronic pneumonia	Confirmed by histopathology of BAL	[115]
** *Multiple EM identified* **				
Not mentioned	Peripheral blood and BM	Differential diagnosis in immunocompromised patients	Histoplasmosis (confirmed by histopathology), talaromycosis	[116]
DNA (ITS region)	FFPE tissue	Retrospective evaluation of the NGS clinical utility	Confirmed by histopathology	[117]

BAL: brochoalveolar lavage; CSF: cerebrospinal fluid; FFPE: formalin-fixed paraffin-embedded; BM: bone marrow.

## Data Availability

Not applicable.

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
