# Peer review of "Molecular Diagnosis of Endemic Mycoses"

_jof, 2022, doi:10.3390/jof9010059_

Round 1

Reviewer 1 Report

Valero et al. show at great length the difficulties and challenges in detecting endemic mycoses, which to some extend holds true for acute infections caused by mycoses. The work of Valero et al. is sound and shows to great detail what work has been achieved throughout. It gives the interested reader a good overview of available references for different pathogenic species and shows the limitations of those methods presented in this manuscript. However, it might, in the interest of completeness, worth looking at NGS and PCR a bit deeper. The authors acknowledge that PCR as well as NGS in specific can serve to revolutionize the way endemic mycoses are detected and thus change the way these events are recorded. With more and more NGS based assays becoming available it would be beneficial to the manuscript to not only have a look into the past but also, given all the ambiguities, which assays might be available in the near to mid term future - if information is openly available. The manuscript would benefit from a conclusion or summary at the end of each paragraph summarising the findings as well as pitfalls and shortcoming of each method presented throughout the specific section which then should lead to an overall summary of suggested next steps and necessary research to even advance further the applicability of molecular methods as well as their acceptance. 

Author Response

Please find below the details of the changes that we have done to our manuscript "Molecular diagnosis of endemic mycoses" according to the reviewer's reports. All changes are highlighted in red throughout the revised manuscript.

Reviewer 1

Valero et al. show at great length the difficulties and challenges in detecting endemic mycoses, which to some extend holds true for acute infections caused by mycoses. The work of Valero et al. is sound and shows to great detail what work has been achieved throughout. It gives the interested reader a good overview of available references for different pathogenic species and shows the limitations of those methods presented in this manuscript. However, it might, in the interest of completeness, worth looking at NGS and PCR a bit deeper. The authors acknowledge that PCR as well as NGS in specific can serve to revolutionize the way endemic mycoses are detected and thus change the way these events are recorded. With more and more NGS based assays becoming available it would be beneficial to the manuscript to not only have a look into the past but also, given all the ambiguities, which assays might be available in the near to mid term future - if information is openly available. The manuscript would benefit from a conclusion or summary at the end of each paragraph summarising the findings as well as pitfalls and shortcoming of each method presented throughout the specific section which then should lead to an overall summary of suggested next steps and necessary research to even advance further the applicability of molecular methods as well as their acceptance. 

We thanks the reviewer for the suggestions. A small concluding paragraph has been included in each part. Regarding NGS, all works have reviewed and, unfortunately, data are limited to date. More studies are need to be able to recommend or suggest methods. A sentence has been added in this sense in the “Next generation sequencing (NGS)” section

Reviewer 2 Report

In this manuscript, authors summarize the existing molecular methods for diagnosis of endemic mycoses, such as histoplasmosis, coccidioidomycosis, paracoccidioidomycosis, blastomycosis, talaromycosis, emergomycosis and lobomycosis; describing their targets and the percentage of sensitivity and specificity for each test. The paper includes a comprehensive and well-organized amount of information, pointing out the relevance of a proper standardization for molecular methods used in diagnostic of these mycoses, aiming to minimize the long-time periods often needed to confirm diagnosis with traditional methods.

Although only 57.6 % of the references consulted were published within the las 5 years, all included references are relevant to the study.

This manuscript addresses a topic that has not received an updated review in recent years, therefore it is relevant to gather the state of the art regarding molecular diagnosis of these mycoses.

Minor concerns:

  • In line 54, “100000” should be expressed as “100.000” to ensure homogeneity with the rest of the manuscript.
  • Although Figure 1 is very descriptive, further explanation of it should be addressed in the text.
  • In section 2.1. authors mentioned that PPK and CFP4 genes were used as targets for PCR in the diagnosis of Histoplasmosis, nonetheless, they are not included in table 1.
  • In line 203 “DNAr” should be “rDNA”
  • Line 300, “year´s” should be “years”
  • In section 2.5 authors do not mention data concerning LAMP, which it is included in Table 1 and appears to have 100% sensitivity and specificity. Elaborating the discussion regarding this technique and the growing trend to expand its use for the advantages it represents in some regions, including those with endemic mycoses would be interesting and improving.
  • In table 1, for some tests, in the column sensitivity/specificity, an “S” appears before some percentages, but it is not described in the table notes (lines 234, 235). The same applied to “PE” appearing in the Blastomycosis section of the same table. In the section of Talaromycosis of the table, the second and fourth rows state “Convetional” instead of “Conventional”.
  • FFPE in Table 1 and 2 has different definitions.
  • In general, Tables are concise, but I suggest limiting the Reference column to a simple citation using the number of reference, instead of also including the name and year, thus allowing to widen the other columns containing much more information to display, for better reading. 

Author Response

Minor concerns:

-In line 54, “100000” should be expressed as “100.000” to ensure homogeneity with the rest of the manuscript.

The change has been done.

-Although Figure 1 is very descriptive, further explanation of it should be addressed in the text.

We thank the reviewer for the suggestion, a new sentence adding information has been included.

-In section 2.1. authors mentioned that PPK and CFP4 genes were used as targets for PCR in the diagnosis of Histoplasmosis, nonetheless, they are not included in table 1.

We thank the reviewer for the suggestion, that article was not included in Table 1 due to the reduced number of clinical samples tested. Now it has been included as reviewer suggested.

-In line 203 “DNAr” should be “rDNA”

The correction has been made.

-Line 300, “year´s” should be “years”

The correction has been made.

-In section 2.5 authors do not mention data concerning LAMP, which it is included in Table 1 and appears to have 100% sensitivity and specificity. Elaborating the discussion regarding this technique and the growing trend to expand its use for the advantages it represents in some regions, including those with endemic mycoses would be interesting and improving.

Done as requested, a sentence has been added in section 2.5 mention the LAMP assay. In addition, a sentence has been added in the conclusion and perspectives section.

-In table 1, for some tests, in the column sensitivity/specificity, an “S” appears before some percentages, but it is not described in the table notes (lines 234, 235). The same applied to “PE” appearing in the Blastomycosis section of the same table. In the section of Talaromycosis of the table, the second and fourth rows state “Convetional” instead of “Conventional”.

Corrected as requested.

-FFPE in Table 1 and 2 has different definitions.

Corrected as requested.

-In general, Tables are concise, but I suggest limiting the Reference column to a simple citation using the number of reference, instead of also including the name and year, thus allowing to widen the other columns containing much more information to display, for better reading. 

We thank the reviewer for this useful comment. Tables have been formatted accordingly.

Reviewer 3 Report

Authors performed a comprehensive review on the molecular diagnosis of endemic mycoses. It is an interesting subject as their diagnosis is difficult and molecular tools are promising.

The review is well written and structured and brings important data for the reader.

I have some minor points you can find in the file attached. The main one is that authors should insist in the differences of performances between sample types. Indeed, as other fungal PCR, sensitivities for endemic mycoses are better on the suspected site of infection, but serum may lack of sensitivity especially in localized forms in immunocompetent patients. I think it is important to advise the reader on this point.

Author Response

Please find below the details of the changes that we have done to our manuscript "Molecular diagnosis of endemic mycoses" according to the reviewer's reports. All changes are highlighted in red throughout the revised manuscript.

Reviewer 3

The review is well written and structured and brings important data for the reader.

I have some minor points you can find in the file attached. The main one is that authors should insist in the differences of performances between sample types. Indeed, as other fungal PCR, sensitivities for endemic mycoses are better on the suspected site of infection, but serum may lack of sensitivity especially in localized forms in immunocompetent patients. I think it is important to advise the reader on this point.

We thank the reviewer for the corrections pointed out in throughout the text and the last suggestion. Although, sensitivity values were not specifically explained in the table, a couple of sentences were included throughout the different sections highlighting those differences. Some corrections have been made trying to increase clarity.

Reviewer 4 Report

In this paper, the authors reviewed the literature about molecular diagnosis of endemic mycoses. The paper is well writen, well organized and provide up-to-date information.

I have only minor comments:

1. In Figure 1, level (high, medium, low) of sensitivity of the different techniques could be added. this is an important factor in the advantages / limitations of each technique.

2. For Paracoccidioidomycosis, line 205 it is stated that sera samples were not recommended. Please add a sentence to explain why serum is not a good sample.

3. Table 1: For better readability, I suggest to split sensitivity and specificity in two different columns.

Author Response

Please find below the details of the changes that we have done to our manuscript "Molecular diagnosis of endemic mycoses" according to the reviewer's reports. All changes are highlighted in red throughout the revised manuscript.

Reviewer 4

In this paper, the authors reviewed the literature about molecular diagnosis of endemic mycoses. The paper is well writen, well organized and provide up-to-date information.

I have only minor comments:

  1. In Figure 1, level (high, medium, low) of sensitivity of the different techniques could be added. this is an important factor in the advantages / limitations of each technique.

    We thank the reviewer for the suggestion, the level of sensitivity has been             added in the   Figure 1

  1. For Paracoccidioidomycosis, line 205 it is stated that sera samples were not recommended. Please add a sentence to explain why serum is not a good sample.

      Done as requested. A sentence has been added explaining the reason.

  1. Table 1: For better readability, I suggest to split sensitivity and specificity in two different columns.

      Done as requested. Sensitivity and Specificity appear now in two columns